# The Household Food Security and Feeding Pattern of Preschool Children in North-Central Nigeria

**DOI:** 10.3390/nu14194112

**Published:** 2022-10-03

**Authors:** Bosede Alice Omachi, Annette Van Onselen, Unathi Kolanisi

**Affiliations:** 1Department of Dietetics and Human Nutrition, University of KwaZulu-Natal, Pietermaritzburg 3201, KwaZulu-Natal, South Africa; 2Department of Human Nutrition and Dietetics, Sefako Makgatho Health Sciences University, GaRankuwa, Pretoria 0208, KwaZulu-Natal, South Africa; 3Department of Consumer Science, University of Zululand, Empangeni 3201, Kwazulu-Natal, South Africa

**Keywords:** preschoolers, food insecure, minimum dietary diversity, feeding pattern, Niger State, Nigeria

## Abstract

Children’s feeding patterns and health outcomes are important determinants of any country’s food and nutrition security status. This study assessed the household food security and feeding patterns of preschoolers in Niger State, Nigeria. A cross-sectional descriptive design and a multi-stage sampling technique were employed to analyze 450 preschool children from selected local government areas. Household food security was measured using the HFIAS nine-item questionnaire, and feeding patterns were evaluated using the qualitative food frequency questionnaire (FFQ). The mean age of the preschoolers was 3.71 ± 0.80 years. A majority (61.30%) of the children consumed cereal-based products, while fruits and vegetables were the least consumed (16.40%). More than half (59.80%) of the preschoolers met their minimum dietary diversity. Almost all (98.80%) of the children were from food-insecure households, with 40.3% being severely food insecure. Parity, religion, and having a breadwinner and source of potable water were significantly associated with the adequacy of minimum dietary diversity (MDD) among the preschoolers (F value = 5.528, *p* ≤ 0.05). The contribution of poor feeding patterns and household food insecurity to the overall health outcome of preschoolers cannot be overlooked. Hence, nations must prioritize improving the availability, accessibility, and utilization of food to better meet the nutritional needs of preschool children.

## 1. Introduction

Africa is ranked second to Asia as the continent with the highest number of malnourished children in the world, with an increase in the number of triple burdens of malnutrition (TBM) (the coexistence of undernutrition, over nutrition, and hidden hunger within the same household) cases [1,2]. 

The prevalence of TBM among children in many low- and middle-income countries (LMICs) due to a compromised dietary intake is alarming and has significantly contributed to the high prevalence of childhood illnesses and mortality in these regions. A global burden of disease study reported that diets lacking adequate nutrients are now the leading cause of death worldwide across all age groups [1].

Similarly, scholars have identified several factors attributed to poor health outcomes and the poor nutritional status of school children, some of which are compromised dietary intake and poor feeding patterns and hygiene practices [3]. Other determinants of poor health among growing children are household food (in)security status, poor socioeconomic status, the wealth index of the caregiver and family heads, and unhealthy food environments at the community and household level across all sociocultural settings in many developing countries [4]. 

The trend and level of childhood malnutrition in the world shows that two out of five under-five children are stunted, and more than one-quarter are either wasted or overweight. In West Africa, 27.7% of under-five children are stunted, 7.5% are wasted, which is higher than the global estimates (21.3%, 6.9% respectively), while about 1.9% of these children are overweight, although this is lower than the global estimate (5.6%). However, this prevalence is an indication that many children consume insufficient healthy foods [5]. 

The consumption of unhealthy food among children such as monotonous starchy staples with limited animal protein and fruits and vegetables, and the overconsumption of sugar-sweetened beverages (SSB) and highly processed fatty and salted snacks, often starts during the complementary feeding period (4–6 months of life) [6,7]. In West Africa, only 34% of children are exclusively breastfed, thus depriving them of good nutrition. Furthermore, during the introduction of complementary feeding, more than three in four children between 6 and 23 months of age did not meet their minimum dietary requirements [1].

Nigeria, the most populous country in Africa, faces an enormous burden of malnutrition among preschool children and thus occupies the second position globally [8,9]. About 14 million under-five children are stunted in the country. This implies that one out of three Nigerian children are malnourished [10], that is, most children are deprived of essential nutrients for optimal development and productivity. Malnutrition also accounts for 33.0% of the under-five mortality rate annually. This prevalence varies across the nation, with higher under-five mortality rates in the northern region [11,12]. Malnutrition among children has been attributed to suboptimal food intake during the developmental stages of the life cycle. For instance, in Nigeria, only 17% of infants are exclusively breastfed, while 18% of children aged 6–23 months are fed with minimum acceptable diets due to maternal and household food insecurity, especially among rural communities [13,14]. The resultant effect is severe acute malnutrition (SAM) which is more pronounced among under-five children, especially in the northeastern and northwestern geopolitical zones of the country, where about 50% of under-five children are stunted [12,13].

The dietary guidelines for Americans recommend that preschool children consume between 1300 and 1800 kcal/day for healthy growth. This should cut across food groups such as grains, fruits, vegetables, milk, meat, and beans. For instance, the intake of about 14.5 to 24 g/day of protein, 1 to 2.5 cup eq/day of vegetables, 1 to 1.5 cup eq/day of fruits 2 to 2.5 cup eq/day of dairy products, and 400 µgRE of vit. A are recommended for under-five children [15]. However, the household food deficit across many households in Nigeria has resulted in inadequate intake among preschool children. 

According to the World Health Organization (WHO) and the Centers for Disease Control and Prevention (CDC), preschool children are between 3 and 5 years old. This period entails intensive cognitive development in children where developmental milestones that help shape their personality, interpersonal relationships, and thinking patterns are fostered; hence, the need to consume healthy meals following the recommended minimum dietary diversity requirements of at least 5 to 8 food groups per day for children is a bedrock for positive health outcomes [16,17,18].

The inadequate intake of healthy food among under-five children in Nigeria is complex and multifactorial, ranging from poor dietary diversity among households, poor wealth index, food illiteracy, high dependency ratio, conflicts, seasonality, and geographical location, among others [14,19,20,21]. Consequently, inadequately fed children, especially preschoolers, may not develop to their full potential both physically and mentally before starting formal education, thus impairing school attendance and intelligence quotients for good academic performance, and eventually leading to poor labor productivity and a reduction in the economic growth of the nation [4,11,22,23,24].

The continuous destruction of lives, communities, and farmlands poses a severe threat to food availability and affordability, resulting in economic shocks, insecurity, and constant hikes in food commodity prices across the nation, which have affected several households in both urban and rural communities across the six geopolitical zones [14,25,26]. Hence, Nigeria faces food accessibility and affordability crises, severely impacting its citizens’ food and nutrition security status [25,27,28,29].

Conflicts have cumulatively affected the food and nutrition security status of the nation because of reduced animal and crop husbandry, and thus increased the risk of hunger and TBM among the vulnerable masses, mostly children across the northern geopolitical zones [25,30,31]. Consequently, this impacts household food security, feeding patterns, and health outcomes of preschool children. Studies have shown that no community or household can sustain healthy food availability if there is an insecurity of lives and property [21,32,33]. 

Moreover, the global economic downturn has affected several countries, and its impact on low-and middle- income countries (LMICs) like Nigeria is evident in the increased poverty rate among the population [14,19,34]. Generally, low incomes, rising food prices, poor policies and governance, growing demand, and changes in consumption patterns not commensurate with production and distribution have implications for households’ food security status [14,25,35,36]. Poverty reduces the affordability of safe and nutritious healthy foods across the different socioeconomic classes, although this impact varies from region to region. Poverty and poor dietary intake are higher in the northern region than in the southern region because of the incessant Boko Haram attacks in the northern region [19,20,21,37].

Recent reports show that about 73 million (about 36.5%) persons in Nigeria cannot afford three consecutive meals per day because of the high poverty rate, national food-deficit, and high cost of available staples [14,34,36,38,39]. Moreover, one out of every three Nigerian children are malnourished and about 44 percent are stunted, 32 percent underweight, and 11 percent wasted [10,11].

Hence, there is a need to create databased awareness of the risk of poor nutrition, hunger, starvation, illnesses, and untimely deaths in the country, especially in badly affected regions. In 2019, about 60% of the disposable income of Nigerians was spent on food expenditure, so many low-to middle-income families, especially those with preschool children, are not meeting their basic needs for a healthy and productive life. This is largely due to suboptimal food intake in both quality and quantity as the country battles insurgency and economic shocks [14,29,39]. Thus, increasing the risk of various forms of malnutrition and food and nutrition insecurity among preschool children [21,40].

Reports on food security and the feeding patterns of children in the northern geopolitical zones are scarce, especially in Niger State, with an increase in the number of triple burdens of malnutrition (TBM) and mortality among under-five children. There is an urgency for national and geopolitical government initiatives and policies that are practicable to combat poor feeding practices and their consequences among preschoolers. Therefore, this study assessed the household food security and feeding patterns of preschoolers in Niger State, Nigeria. 

## 2. Materials and Methods

### 2.1. Target Population

In this cross-sectional descriptive study, a total of 450 preschool children in Niger State, one of the states in the northcentral zone of Nigeria, were selected by the multi-stage cluster sampling method. There are three senatorial districts in the state, namely, Niger east, Niger north, and Niger south. For this study, Niger south was selected by purposive sampling because it is one of the major farming communities not enlisted among the flash points of Boko Haram terrorist attacks in the state. Simple random sampling was used to select five local government areas (LGAs) out of the eight LGAs. 

Systematic random sampling was used to select five political wards each and their primary healthcare facilities based on the national population records available at the five selected local government area headquarters. The total number of households was also obtained. Eighteen households with preschool children based on the national population census (NPC) record in each political ward across the five LGAs were randomly selected for this study. 

### 2.2. Sample Size

The minimum sample size for the study was estimated using the formula established by Cochran and colleagues for population-based cross-sectional studies n = z^2^p(q)d^2^ [41], where n is the sample size to be determined, z is the z-score of 1.96 at a 95% confidence level, p is the estimated proportion of an attribute that was present in the population (48.0%, the proportion of the food insecure population in rural households in North-Central Nigeria from previous study) [42], d is the desired level of precision 5%, and q is 1 – p, considering a 10% nonresponse rate and design effect of 2% among the preschool mothers.

### 2.3. Procedures

Household food security was measured using the HFIAS nine-item questionnaire associated with the experience of food insecurity within the household over 30 days. Before the survey, a pilot study was conducted to review the questionnaire to adapt the phrases and definitions to the local context by using semi-structured in-depth interviews. 

Trained research assistants filled in the questionnaires during door-to-door interviews with the mothers and their preschool children. Based on the HFIAS questionnaire scores, households were grouped into four categories of food access (in)security: secure (0–1), mildly food insecure (2–7), moderately food insecure (8–14), and severely food insecure (15–27). Supplementary questions concerning meal frequency and preference were evaluated using the qualitative food frequency questionnaire (FFQ), while the demographics and socio-economic characteristics of the households, mothers, and their preschool children were also completed in the interviews to assess the household food preference, feeding patterns, biodata, and anthropometric indices of the preschoolers.

### 2.4. Statistical Analysis of the Results

Data obtained were coded and analyzed with SPSS version 27. Descriptive analyses were presented in frequency and percentages, chi-square was used to determine the statistical significance of the dependent variables, while a linear regression model was used for inferential statistics at *p* ≤ 0.05.

## 3. Results and Discussion

Four hundred and fifty preschool children were recruited for this study. The socioeconomic and demographic characteristics of the preschool children and their mothers are listed in Table 1 and Table 2. The majority (55.0%) of the preschool children were males, while 45% were females; this finding is consistent with a similar study conducted in Benue State by Omotesho et al. [10] where more than half (55.4%) of the children were males. It is also consistent with another study completed in Southwest Nigeria by Akanbiemu et al. [43] where 53.3% of the children were males. The male predominance observed in the current study is in contrast with a South African study by Chakona [44] where more than half (56.0%) of the children were females. The mean age of the children is 3.71 ± 0.80 years, which is higher than the 1 year and few months recorded by Omotesho et al. [10] and Chakona, respectively, [44] but similar to the findings of Akanbiemu et al. [43] where the mean age of the children was 4.3 ± 1.3 years. 

More than two-thirds (76.4%) of the preschool children and their mothers were from rural areas, which is not in agreement with a study conducted in Ethiopia, where about two-thirds of the mothers resided in urban centers [45]. This could be because the study participants were predominantly from the rural area of the state. Almost all (95.6%) of the children came from households that practice Islam, which is similar to the findings of Ahmed et al. [46], where about two-thirds of the mothers and their children practiced Islam as a religion. This could be because the study area and Niger State at large are Muslim-dominated. However, in Tegedie District, northeastern Ethiopia, most (85.3%) of the mothers belonged to the orthodox religion [45].

The majority (73.6%) of the preschool children came from households where fathers were the breadwinners, and 42.7% had casual farm labor as their occupation, with a monthly income less than $100 (N33,000), which is similar to the report of Akanbiemu et al. [43] where the average household per capital income was $43.4 (<N18,000). However. a study conducted in southwest Nigeria, in contrast, reported that 38.6% of the children’s fathers were traders and the majority (54.8%) of them earned less than $48.2 (N20,000) monthly [47]. The poor household income in the study area could be because many fathers are either small-scale farmers or menial workers who are often poorly paid. This explains why most of the children were from food-insecure households, because income contributes significantly to the food and nutrition security status of the vulnerable members of a household. The mothers in the study were predominantly unemployed housewives (71.6%), which is similar to the reports by Teshome et al. [45] and Ahmed et al. [46] where 57.3% and 46.9% of the mothers, respectively, were housewives and unemployed. The unemployment status and poor income of mothers and caregivers are social determinants that often negatively impact the available disposable income needed to acquire safe and nutritious food, especially among households with preschool children in food-insecure communities.

Less than half (45.6%) of the mothers of these preschool children never had any formal education besides Arabic education; this finding is similar to the study conducted in Tegedie District, northeastern Ethiopia, where about two-third (60.3%) of the mothers had neither formal education nor Arabic education [45]. In contrast, Ahmed et al. [46] and Akanbiemu et al. [43] reported that almost half (45.4% and 42.8%) of mothers had primary school and secondary education, respectively, as their highest educational status. The poor educational status of mothers in the current study could be attributed to early marriage and childbirth, which is a common practice in the northern region of Nigeria. In addition, religious and cultural beliefs among the northerners that girl-child education is a waste of resources are also a factor. This poor disposition to female education contributes to the poor feeding practices and dietary patterns among preschool children since quality education improves income status and empowers women to make healthy food choices, purchase sufficiently safe and healthy foods, and adopt good nutritional practices that enhance positive health outcomes and food security among children [14,21,48]. In addition, the educational status of the breadwinner, caregiver, and household head has been associated with household food security status, dietary diversity, feeding patterns, and the anthropometric status of the preschoolers [38,49,50]. This is consistent with other findings which suggest that educated household heads or caregivers are more aware of the possible advantages of healthy food habits and eating practices [48]. Similarly, food literacy enables mothers to make informed decisions on food acquisition, meal plans, and processing and preparation to ensure the optimal growth and development of their children [50]. Education and food literacy also enhance the households and individual food and nutrition security, and eventually, favorably impact the academic performance of children, labor productivity in adulthood, gross domestic product (GDP) of the nation [51,52].

Almost all (98.2%) of the mothers were married, which agrees with the findings of Akanbiemu et al. [43], Teshome et al. [45], and Ahmed et al. [46], where 93.8%, 90.7%, and 92.5% of the participants, respectively, were married. However, this contrasts with a study conducted among farming households in southwest Nigeria where only 60% of the participants were married [14]. The high proportion of children from legally married families in the study area could be because Islam frowns on premarital affairs and childbirth outside wedlock. 

The majority (70.9%) of the children were from multiparous mothers (2–4 children), which is consistent with the findings of Karcz et al. [53] and Ahmed et al. [46] where more than 80% and 65.7% of the participants were multiparous mothers. High parity among low-income-earning households contributes to food insecurity and poor feeding patterns among younger children because parity is a strong predictor of the quality of the preschool children’s diet [40,54]. Owoo [21] and Ware et al. [55] reported that the number of growing children in a household affects the quantity and quality of meals available per capital head consumption for optimal growth and development, thus increasing their vulnerability to food and nutrition insecurity since available resources are spread more thinly across the dependent members of the households.

The prevalence of food security among preschool children in Niger State is shown in Figure 1. The majority (98.9%) of the households were food-insecure, ranging from mildly (16.2%), moderate (42.4), to severely food insecure (40.3%), while few (1.1%) were food secure. This is in tandem with the findings of Yahaya et al. [47], Akanbiemu et al. [43], and Omotayo et al. [14] where 63%, 80.9%, and 78% of children, respectively, in a study conducted in southwest Nigeria were food insecure. A similar report by Macharia et al. [56] also showed that most children were from food-insecure households. Although, the report by Yahaya et al. [47] showed a higher prevalence (36.0%) of food-secure children, this could be because of the high maternal educational status, women empowerment, and low parity obtainable in the southern region of Nigeria as compared with the northern region. There is also variation in the prevalence of food security among urban and rural dwellers as opined by Akanbiemu et al. [43]. This corroborates the fact that the prevalence of food insecurity in the country varies across the geopolitical zones and places of residence and therefore is higher in the northern region than in the southern region and also among rural areas than urban areas [19,21]. Most of the mothers of the preschool children admitted that they had experienced food insecurity or worry of not having food to eat due to lack of money and food deficits, and some said their households ate nothing some days due to lack of income to acquire food (Table 3), which is similar to the report by Yahaya et al. [47] where 24.8% and 7.8% were food insecure with moderate-to-severe hunger, respectively.

However, Etana and Tolossa [57] reported a similarly low trend of food security (12.4%) among the urban participants in Ethiopia. The high prevalence of food insecurity in the study area must have accounted for regarding why many preschoolers could not meet their minimum dietary diversity (MDD) and minimum meal frequency (MMF). The high prevalence could be attributable to the poor educational status, unemployment status, insufficient income, and low financial empowerment (high poverty level) experienced by the mothers, as opined by [34,38,40,57]. Similarly, most mothers indicated that their household experienced various forms of anxiety of food sufficiency and hunger due to a lack of resources to access sufficient food to meet their needs adequately or due to food shortages within the previous month, as shown in Table 3.

Moreso, Omotesho et al. [10] Ezeh et al. [12] Kandeepan et al. [58] and Sani Nass et al. [13] opined that household food insecurity contribute significantly to severe acute malnutrition and micronutrient deficiency disorders among under-five children, while Ahmed et al. [46] found that the household’s food security status affected the quality of complementary food of 6–23-month-old children.

The feeding pattern of the preschool children over a 7-day recall is indicated in Figure 2. The food frequency consumption pattern showed that cereal was the most consumed, followed by root and tubers among the preschool children; this is similar to the findings of Ahmed et al. [46] who reported the same trend of food consumption among 6–23-month-old Ethiopian children. Cereal food was most consumed about three to six times a week (61.4%). 

Most of the mothers indicated that the starchy staples such as cereals and root and tubers were the most preferred food by the households and thus the most consumed by the preschoolers, (Figure 3). This could be because cereal crops such as rice, millet, and maize are the most cultivated staple food in the study area and thus the most affordable. In the interview held with some of the children, starchy staples such as rice-based and millet-based meals were identified as frequently consumed food groups on a daily basis, which could be because of cultural food preferences and the type of food commonly cultivated among farming communities in Niger State. This is in tandem with the findings of Ahmed et al. [46] who reported a similar food consumption pattern among children in Shashemene Ethiopia. Reddy and Dam [59] affirmed that culture and identity play a vital role in food selection, acquisition, consumption, and ultimately nutritional status in any society, which is in contrast with a similar study conducted in Uganda by Nabuuma et al. [60] where white roots and tubers with bananas were the most consumed food among children. However, Niger State is one of the major producers of rice and millet in Nigeria.

Fruit and vegetable consumption was very insufficient among the preschoolers. Webb and Lewis [61], Grimm et al. [62], Duran et al. [63], Reddy and Dam [59], Woodside et al. [64], and Ahmed et al. [46] also indicated the same trend of low fruit and vegetable consumption among children. The low consumption in the study area could be attributed to factors such as poor nutrition knowledge of the health benefits of fruits and vegetables and the impact of seasonality on affordability, because most mothers during the interview mentioned that “we consume fruits only when they are readily available since we pick them from the neighborhood, others indicated consumption only during Ramadan fast’ when asked the reason for consumption, the children mention’ to quench hunger pangs’ indicating that many homes do not purchase fruits for consumption but rather engage in wild harvesting (pluck from trees) during gluts. Therefore, these responses showed that there is poor maternal knowledge on the importance of fruits and vegetables to health, which explains why the children seldomly consume them in the study area, especially among the rural dwellers and low-income households.

The number of children who met their minimum dietary diversity (MDD) and MMF was low compared with those not meeting theirs, as indicated in Figure 4, which is not consistent with the findings of Ahmed et al. [46], who reported that the majority (61.7%) of the children met their MMF, while 42.5% met their MDD.

A study by Macharia et al. [56] reported that although infants’ feeding practices were inappropriate, most of the children were able to meet their MMF. Similarly, the household food security status and minimum meal frequency (MMF) of consumption among the preschoolers were significantly associated with the adequacy of their minimum dietary diversity (MDD) (*p* < 0.05), as shown in Table 4, Table 5 and Table 6, which is similar to the trend reported by Ahmed et al. [46], where the food security status of households influenced the (MDD) and meal frequency (MMF) of the children.

In this study parity, religion, household breadwinner, and portable water source for household use were strong predictors of a child’s minimum dietary diversity adequacy (F value = 5.528, *p* < 0.000) as indicated in Table 7. In a similar study conducted by Balistreri [65] and Ware et al. [55], parity and family size were significantly associated with children’s food security status. Other factors reported to significantly affect the adequacy of under-five children’s dietary intake in both quality and quantity among Nigerians as reported by Akanbiemu et al. [3,43], Fadare et al. [38], Owoo [21,27], Obayelu and Akpan [19], Yahaya et al. [47], and Omotayo et al. [14] were household socioeconomic status, maternal/caregiver food literacy, geopolitical location, family structure and high dependency ratio, seasonality, conflicts, affordability due to hike in food prices, poor policy, and bad governance.

## 4. Conclusions

This study has shown that children’s feeding patterns, household socioeconomic status, and dietary diversity are important indicators of food and nutrition security status among preschool children from low- and middle-income households in Niger State, Nigeria. Many of the preschool children could not meet their minimum dietary intake of between 5 to 8 food groups and MMF because they were majorly from food-insecure households. The poor dietary diversity among preschool children was attributed to the food insecurity and poor socioeconomic status of the household. Most of their mothers had poor educational status and were unemployed, and thus were financially incapacitated to provide adequately for the preschool children’s dietary needs. The frequency of consumption of food groups such as root and tubers, eggs, snacks, and Vit.A-rich fruits and vegetables were significantly associated with the food security status of the preschool children. At the same time, socioeconomic factors such as parity, religion, breadwinner, and source of potable water for household use were predictors of minimum dietary diversity of the preschoolers. Poor income, parity, and food deficits depleted the available resources in the household and overstretched the meager disposable income available to afford sufficiently safe, nutritious food for healthy living for preschool children. Some of the factors that could contribute to the low consumption of animal protein, fruits, and vegetables among the respondents are hikes in food prices attributed to insecurity challenges faced by these farming communities, thereby compromising the dietary quality of growing children. Apparently, compromised feeding patterns of preschool children could be detrimental to the overall health and nutrition security status of under-five children in Nigeria and many developing countries. Therefore, stakeholders and policymakers should prioritize improving the availability, accessibility, and utilization of nutritious food to better meet the nutritional needs of preschool children, especially among resource-limited communities. In addition, strategies geared towards improving the security situation of the country and fostering food production must be adopted. Investment in nutrition-sensitive programs such as value-adding technology along the food chain system to enhance the availability and affordability of nutrient-dense staple foods among rural dwellers must be explored. Mothers should be enlightened on the health and economic benefits of home gardening and small-scale animal husbandry in improving dietary diversity among preschool children from poor households, thereby ameliorating the food and nutrition insecurity, childhood morbidity, and mortality prevalent in rural communities in many developing countries such as Nigeria. 

## 5. Limitation of the Study

This study’s inability to quantify the preschool children’s meals using a 24 h dietary recall is a major limitation because at the time this study was conducted, Nigeria did not have a food composition database. Secondly, the study did not cover children from communities under the threat of Boko Haram attacks. However, further studies should explore the possibility of quantifying the essential nutrients consumed by the preschoolers in Niger State with regards to the recommended nutrient intake (RNI), estimated energy requirement (EER), estimated average requirement (EAR), recommended dietary allowance (RDA), adequate intake (AI), reference intake range for macronutrients (RIs), and tolerable upper intake level (UL) appropriate for this age group. 

## Figures and Tables

**Figure 1 nutrients-14-04112-f001:**
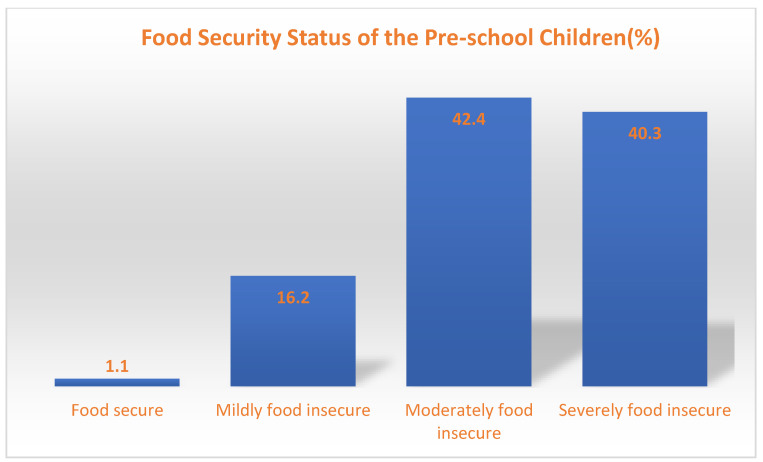
The prevalence of food security among the preschool children in Niger State, Nigeria.

**Figure 2 nutrients-14-04112-f002:**
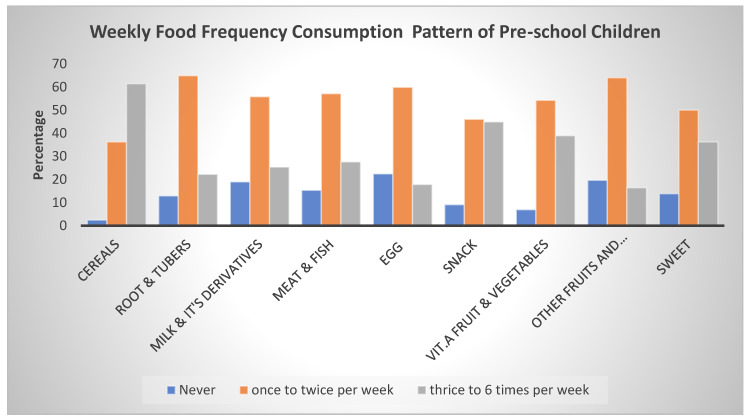
Weekly food consumption patterns of the preschool children in Niger State, Nigeria.

**Figure 3 nutrients-14-04112-f003:**
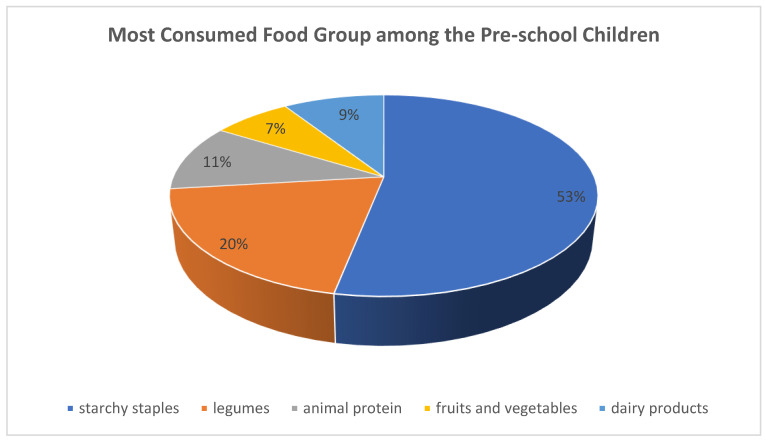
Most consumed food group by the preschool children in Niger State, Nigeria.

**Figure 4 nutrients-14-04112-f004:**
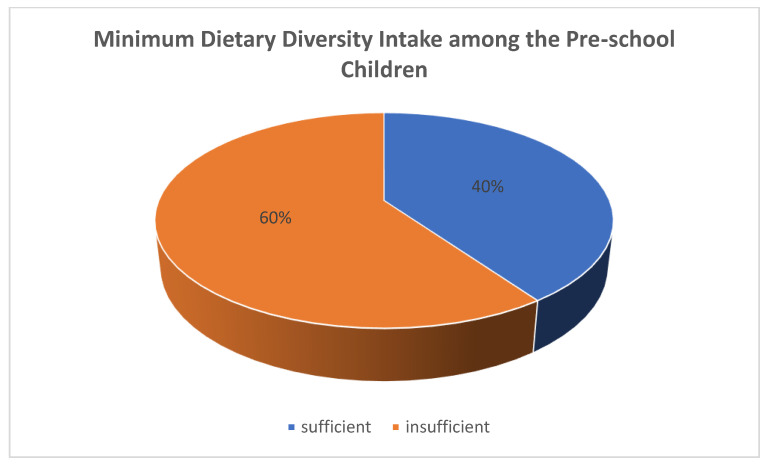
Proportion of preschool children meeting their minimum dietary diversity (MDD).

**Table 1 nutrients-14-04112-t001:** Sociodemographic characteristics of the preschool children in Niger State.

Variable	Frequency (*n* = 450)	Percentage (%)
Residential Area		
Urban Rural	106 344	23.6 76.4
Sex		
Female Male	248 202	55.0 45.0
Ethnic Group		
Nupe Gwarri Hausa Yoruba Others	308 04 113 15 10	68.4 0.9 25.1 3.3 2.3
Religion		
Christianity Islamic Traditional	16 430 4	3.6 95.6 0.8
Breadwinner		
Father only Mother only Both parents Grandparents	331 13 104 02	73.6 2.9 23.1 0.4
Method of water treatment		
Boiling Use of water guard Allowing water to settle/decant None	108 36 32 274	24.0 8.0 7.1 60.9

**Table 2 nutrients-14-04112-t002:** Sociodemographic characteristics of the mothers of the preschool children.

Variable	Frequency (*n* = 450)	Percentage (%)
Maternal age group		
15–25 years 26–35 years 36–45years >46 years	89 287 67 07	19.8 63.8 14.9 1.5
Educational status		
Islamic school/no formal education Primary school only Secondary school only Tertiary education	233 83 96 38	51.8 18.4 21.3 8.5
Marital status		
Married Single	442 08	98.2 1.8
Employment status		
Actively employed Retired Unemployed Casual worker	92 21 322 15	20.4 4.7 71.6 3.3
Parity		
Primiparous Multiparous Grand multiparous	33 286 131	7.3 63.6 29.1
Occupation		
Civil servant Artisan Full housewives Self-employed/petty trading Farming	20 19 123 206 82	4.4 4.2 27.4 45.8 18.2
Income range (N)		
<N33,000 N33,000–N50,000 N51,000–N100,000 N101,000–N150,000	377 62 05 06	83.8 13.8 1.1 1.3

**Table 3 nutrients-14-04112-t003:** Household Food Insecurity Access Score (HFIAS) of the preschool children.

HFIAS Questions	NO		Yes						Total	
			How Often Did This Happen in a Month?							
			Rarely		Sometimes		Often			
	Freq	%	Freq	%	Freq	%	Freq	%	Freq	%
1. Did you worry that your household would not have enough food?	163	36.6	85	18.5	77	17.1	125	27.8	450	100
2. Was anyone unable to eat a preferred food because of lack of resources?	119	26.4	90	20.0	108	24.0	133	29.6	450	100
3. Did anyone eat a limited variety of food due to lack of resources?	130	28.9	105	23.3	105	23.3	110	24.5	450	100
4. Did anyone eat what they did not want because of lack of resources?	148	32.9	120	26.7	84	18.7	98	21.7	450	100
5. Did anyone eat less than needed because of the lack of enough food?	112	24.9	104	23.1	137	30.4	97	21.6	450	100
6. Did anyone eat fewer meals per day because of insufficient food?	92	20.4	71	15.8	131	29.1	156	34.7	450	100
7. Was there no food in the house because of lack of resources?	96	21.3	84	18.7	111	24.7	159	35.3	450	100
8. Did anyone go to bed without food?	143	31.8	40	8.9	117	26.0	150	33.3	450	100
9. Did anyone go without food throughout the day and night without any food?	202	44.9	45	10.0	169	37.6	34	7.5	450	100

**Table 4 nutrients-14-04112-t004:** Minimum dietary diversity intake vs. weekly food frequency consumption patterns of preschoolers.

Minimum Dietary Diversity Score	Weekly Food Frequency Intake of Food Groups by the Preschool Children				
	Never	Once to Twice Per Week	Three to Six Times Per Week	Chi-Square	*p*-Value
		Freq (%)	Freq (%)		
Insufficient	-	207 (76.9)	62 (23.1)	32.38	0.000
Sufficient	-	169 (93.4)	12 (6.6)		

Insufficient—intake of 0 to 4 food groups; sufficient—intake of 5 to 8 food groups; never—ate nothing; once to twice—eating at least 1 to 4 food groups in the week; thrice to six—ate between 5 to 8 food groups in the week.

**Table 5 nutrients-14-04112-t005:** Minimum dietary diversity score and household food security status of the preschool children.

Minimum Dietary Diversity Score	Food Secure	Mildly Food Insecure	Moderately Food Insecure	Severely Food Insecure	Chi-Square	*p*-Value
	Freq (%)	Freq (%)	Freq (%)	Freq (%)		
Insufficient	1 (0.56)	20 (11.05)	66 (36.46)	94 (51.93)	18.721	0.000
Sufficient	4 (1.49)	53 (19.70)	125 (46.47)	87 (32.34)		

Insufficient—intake of 0 to 4 food groups; sufficient—intake of 5 to 8 food groups; never—ate nothing; once to twice—eating at least 1 to 4 food groups in the week; thrice to six—ate between 5 to 8 food groups in the week.

**Table 6 nutrients-14-04112-t006:** Weekly food frequency consumption patterns of the preschoolers and their food security status.

Food Groups	Frequency of Weekly Food Consumption	Household Food Security Status				F value	Chi-Square	*p*-Value
		Food Secure	Mildly Food Insecure	Moderately Food Insecure	Severely Food Insecure			
		Freq (%)	Freq (%)	Freq (%)	Freq (%)			
Cereals	Never	-	2 (0.44)	-	9 (2.0)	4.38	17.88	0.007
	Once to twice per week	2 (0.44)	16 (3.56)	79 (17.56)	66 (14.67)			
	Thrice to six times per week	3 (0.67)	55 (12.22)	112 (24.89)	106 (23.55)			
Roots and tubers	Never	2 (0.44)	12 (2.68)	19 (4.22)	25 (5.57)		23.39	0.001
	Once to twice per week	1 (0.22)	33 (7.33)	135 (30.0)	123 (27.33)			
	Thrice to six times per week	2 (0.44)	28 (6.22)	37 (8.22)	33 (7.33)			
Egg consumption	Never	3 (0.67)	13(2.89)	41 (9.12)	44 (9.78)		21.31	0.002
	Once to twice per week	1 (0.22)	36 (8.0)	127 (28.22)	105 (23.33)			
	Thrice to six times per week	1 (0.22)	24 (5.33)	23 (5.11)	32 (7.11)			
Vit. A rich fruits and vegetables	Never	2 (0.44)	10 (2.22)	9 (2.0)	10 (2.22)		22.32	0.001
	Once to twice per week	2 (0.44)	39 (8.68)	116 (25.78)	87 (19.33)			
	Thrice to six times per week	1 (0.22)	24 (5.33)	66 (14.67)	84 (18.67)			
Snack consumption	Never	2 (0.44)	9 (2.0)	10 (2.22)	20 (4.44)		22.58	0.001
	Once to twice per week	2 (0.44)	21 (4.68)	104 (23.11)	80 (17.78)			
	Thrice to six times per week	1 (0.22)	43 (9.56)	77 (17.11)	81 (18.0)			

**Table 7 nutrients-14-04112-t007:** Regression coefficient of the socioeconomic status of the household as predictors of adequacy of minimum dietary diversity of the preschool children.

Predictors	Unstandardized β	*p*-Value	F Value
Parity	0.0175	0.000	5.528
Religion	−0.285	0.017	
Household breadwinner	−0.062	0.021	
Source of portable water for household	−0. 046	0.039	

## Data Availability

Dataset for this study will be made available by the corresponding author on reasonable request.

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
