# Peer review of "The Household Food Security and Feeding Pattern of Preschool Children in North-Central Nigeria"

_nutrients, 2022, doi:10.3390/nu14194112_

Round 1

Reviewer 1 Report

The manuscript presented for review covers an important topic, but requires corrections to be suitable for publication:

- the text requires many editorial corrections - it contains many minor errors that are difficult to cite due to the lack of line numbering, but these errors affect the reception of the manuscript, for example:

- please check what the correct p-value should be in Nutrients - e.g. in italics,

- in the abstract, the last few words are in a different verse, they are detached from the text,

- there is no space in front of the literature in brackets, sometimes the literature is in incorrect brackets, e.g. ([33.37] [38] [19.35]), disordered,

- literature number 65 is empty - should this item be?

- the title is misleading as the reader may get the impression that the authors will investigate safety aspects (e.g. food contamination),

- the authors inserted the text into the template from 2021,

- the methodology should include a separate point: statistical analysis of the results,

- the description of the results and the discussion should be separate,

- tables should be numbered sequentially, not a and b,

- the formatting of tables should be in accordance with the journal's guidelines,

- please check the correctness of the numbers in the tables, e.g. the sum of the percentages in the "ethnic group" is 99.9%, and in the "Maternal age group": 99.7%,

- it does not seem appropriate to discuss the age and origin of respondents in the context in which it has been presented,

- what does Table 2 show? Why are the % total in each question not the same?

- the wording "economically poductive" is not very fortunate when it comes to describing breastfeeding,

- the entire text should be checked in terms of correct formatting, etc.

Author Response

The author's response to reviewer 1's comment has been uploaded.

Reviewer 2 Report

1. This is a very interesting study. The authors should be clear when they write about the sample size. Did they survey households (n=450, p. 3) or children (n=450, p. 4)?

2. The statement "The consumption of unhealthy food among children often..." (introduction part) should be developed. The question is what are unhealthy foods for infants and pre-schoolers?

3. The question is why only 17% of infants are xclusively breastfed in Nigeria? Please, give the readers the reason for it and cite an adequate literature. 

4. The authors should decide whether they use "preschoolers" or "pre-schoolers" and use one of the terms through the whole text.

5. I am not sure about the world "populace". Shouldn't it be "population"? (page 3)

6. When there is a number 73 million persons, the reader is curious what is the share of the population which could noy afford three meals a day. (page 3)

7. The citation in 2.2 Sample size is needed (regarding the formula established by Cochran and colleague(s?) 

8. Please check the number $200 and $112.3 income per day respectively (page 7).

9. Please check the language here: "opined a contrary view" (page 7), "this is in tandem with" (page 8)

10. Figure 1: please replace "in the Study Area" with " in Niger State, Nigeria" (the title); remove "axis title"; Table 2: "...in Nigeria" in the title is needed.

11. Please consider showing the data from table 2 on a figure. It would be more readible.

12. Figure 2: Please insert the following changes - eggs instead of egg consumption, snacks instead of snack consumtion, sweets instead of sweat consumtion.

13. The conclusion is too short. It might be because of the fact that no research question/hypothesis  was stated.

I would suggest considering these issues.

I haven't notice any limitations of this study mentioned, neither the topic of further studies nor suggestions for public/private policy regarding food insecurity of pre-schoolers in Nigera.

I suggest to write about the (possible) reasons of the situation and (possible) ways of improving the state of food security in Nigeria.

Author Response

The response to reviewer 2's comment has been uploaded

Round 2

Reviewer 2 Report

I am satisfied with the revised version of the manuscript. Please consider replacing the statement about "further studies..." from limitations part.